# The Relation of Alpha Asymmetry to Physical Activity Duration and Intensity

**DOI:** 10.3390/brainsci15121322

**Published:** 2025-12-11

**Authors:** Bryan Montero-Herrera, Megan M. O’Brokta, Praveen A. Pasupathi, Eric S. Drollette

**Affiliations:** Department of Kinesiology, University of North Carolina at Greensboro, Greensboro, NC 27412, USA; bgmonterohe@uncg.edu (B.M.-H.); mmobrokta@uncg.edu (M.M.O.); papasupathi@uncg.edu (P.A.P.)

**Keywords:** alpha asymmetry, EEG, affect, accelerometry, GGIR

## Abstract

**Background/Objectives:** Regular physical activity (PA) benefits mood and cognition, yet the neural markers associated with free-living PA remain unclear. Alpha asymmetry (AA), a neural marker of affective and motivational states, may help predict individuals’ preferred activity intensity and duration. To examine the relationship between resting-state AA in frontal and parietal regions, positive affect, and accelerometer-derived PA metrics were measured. **Methods:** Fifty-nine participants (age = 21.8 years) wore wrist accelerometers for 7 days, completed resting-state electroencephalography (EEG; alpha power 8–13 Hz), and completed the Positive and Negative Affect Schedule (PANAS). PA metrics included sedentary time (ST), light PA (LPA), moderate-to-vigorous PA (MVPA), average acceleration (AvAcc), intensity gradient (IG), and the most active X minutes (M2–M120). Multiple regression models tested AA to PA associations while accounting for sex and positive affect. **Results:** Although frontal AA was included as a key neural candidate, the observed associations emerged only at parietal sites. Greater right parietal AA power was associated with the most active M60, M30, M15, M10, and M5. For IG, greater AA power was observed in the left parietal region. No significant associations emerged for LPA, MVPA, AvAcc, M120, or M2. Across models, higher positive affect consistently predicted greater PA engagement. **Conclusions:** While resting frontal AA is theoretically relevant to motivational processes, the findings indicate that parietal AA more strongly differentiates individuals’ tendencies toward specific PA intensities and durations. Positive affect is associated with PA engagement. These findings identify parietal AA as a promising neural correlate for tailoring PA strategies aimed at sustaining active lifestyles.

## 1. Introduction

Regular physical activity (PA) has positive health benefits for physical, mental, and cognitive health, yet most U.S. adults fail to meet public-health guidelines [1,2,3], with approximately 80% classified as physically inactive [2]. To promote sustained PA engagement, researchers have increasingly sought to identify the affective and neural mechanisms that drive individuals to initiate and maintain PA behavior [4,5]. Although numerous studies have demonstrated that PA enhances mood and emotion regulation [6,7,8], less is known about how individual differences in neural activity and affect predict real-world PA behaviors, including aspects of PA intensity and duration. Previous evidence, predominantly in older adults, has demonstrated that specific changes in the structure and function of brain regions are linked to greater PA adherence [9,10]. Similarly, Williams et al. [11] demonstrated that higher positive affect during walking was related to increments in PA at 6 and 12 months. Yet the neural markers underlying these behavioral tendencies remain poorly understood, particularly among younger adults. Identifying neural markers—such as alpha asymmetry (AA)—that capture affective and motivational predispositions toward PA may be a critical step toward explaining why some individuals are more inclined to adopt active lifestyles than others. The present study addresses this gap by examining whether resting AA, measured in both frontal and parietal regions, is associated with free-living PA (i.e., time spent at different intensities per day in natural, unsupervised, and habitual conditions) behaviors across varying intensity levels and durations.

Resting AA, a trait-like electroencephalography (EEG) index linked to approach-avoidance motivation and affective valence, offers a promising lens through which to examine these relationships [12,13,14], given robust relations with PA [15,16,17]. Measured as hemispheric differences in alpha power (8–13 Hz), AA reflects relative cortical activation, where reduced alpha power in a hemisphere indicates greater activation [18]. Frontal AA patterns are consistently associated with motivational and affective states, low right frontal alpha power (i.e., greater activation) is correlated with reduced motivation and heightened stress, and low left frontal alpha power (i.e., greater activation) predicts approach behaviors, positive affect, and resilience [12,19,20,21,22,23]. Because sustained PA requires initiating and maintaining effortful behavior, neural systems supporting approach motivation and positive affect are critical; individuals with leftward AA shifts may find PA more rewarding and less aversive, facilitating adherence over time. Aerobic PA is reliably associated with leftward frontal AA shifts, which predict post-PA mood enhancement [17,23,24,25]. Critically, these effects are modulated by PA parameters—sessions of 30 min at ≥70% VO_2_ max elicit the most pronounced frontal AA changes [26]—though such protocols may only be feasible for highly active individuals [17,19,20,21]. Despite these insights, important gaps remain. Most AA research has focused narrowly on frontal regions, potentially overlooking contributions from parietal AA, which has been linked to arousal, motivation, and motor cortex activation [27,28,29,30,31]. Moreover, recent studies evaluating the relation of AA to free-living PA [32,33] have relied on self-report PA (prone to recall bias, [34,35]) and omitted affective and multi-regional AA measures. The present study aims to address these gaps by integrating accelerometry-derived PA metrics with multi-region AA (i.e., frontal and parietal) and affective measures—an approach that could clarify how neural and psychological mechanisms relate to free-living PA behavior.

While AA has traditionally been investigated over frontal sites, growing evidence suggests that parietal regions also contribute critically to affective and behavioral regulation [36,37,38]. Parietal AA has been associated with arousal, attentional engagement, and sensorimotor integration—processes that may influence one’s readiness to initiate and sustain movement behaviors [39]. Considering these neural substrates, examining both frontal and parietal AA may provide a more comprehensive understanding of how motivational and affective tendencies are reflected in neural activity patterns. This broader perspective sets the foundation for investigating how AA relates to PA.

Accelerometers provide objective, bias-free measurements of PA by capturing movement over discrete time intervals [40]. These devices are widely employed in large-scale studies [41,42,43] due to their ability to quantify PA volume and intensity—including sedentary time (ST), light-intensity PA (LPA), moderate-intensity PA (MPA), moderate-to-vigorous PA (MVPA), and total activity counts per day—across free-living conditions [40]. However, these traditional intensity cut-points are limited by calibration biases and population-specific validation [44], prompting a shift toward 24 h movement analysis [45,46,47]. To address these limitations, newer analytical metrics—average acceleration (AvAcc), intensity gradient (IG), and the most active X minutes (MX metrics)—have been developed to provide a more comprehensive understanding of how people accumulate movement throughout the day.

AvAcc represents the overall magnitude of movement across a 24 h period, reflecting total PA volume regardless of intensity [44]. Individuals with higher AvAcc values engage in greater overall movement, whether through frequent light activity, structured exercise, or a combination of both, making it a global indicator of daily activity exposure. IG represents the negative curvilinear relationship between activity intensity and the time accumulated at each intensity level. A higher IG (i.e., less negative value) indicates greater time spent across the full intensity spectrum, including higher intensities, whereas a lower IG (i.e., more negative value) reflects proportionally less time in moderate-to-vigorous intensities and more time in lower-intensity activity [44,48]. In practical terms, IG distinguishes individuals who accumulate most of their movement at lower intensities (e.g., sedentary or light PA) from those who distribute effort more evenly or reach higher intensities during the day. MX metrics (e.g., M15, M120) identify the most active accumulated minutes within a 24 h period, integrating both volume and intensity [44,49,50]. Each MX value represents the intensity of an individual’s movement during their most active periods—for example, M15 corresponds to the activity intensity during the most active 15 min within a 24 h period. These metrics capture whether a person’s daily activity profile is driven by brief, high-intensity bouts or by longer periods of moderate activity, regardless of whether these minutes occur continuously or are accumulated sporadically across the day. Together, AvAcc, IG, and MX metrics provide complementary insights into total activity exposure, intensity distribution, and peak effort, offering a richer depiction of real-world PA behavior than traditional cut-point methods.

Despite compelling evidence, the field lacks multi-regional AA investigations and studies that integrate affective, neural, and objective PA indicators; most prior work has relied on frontal AA alone and self-reported PA, limiting mechanistic understanding. The present study extends prior work by employing accelerometer-derived measures that more precisely quantify free-living PA. Using these metrics enables continuous assessment of both PA volume and intensity, overcoming the limitations of traditional cut-point methods and self-reported activity. Incorporating these objective measures alongside affective (e.g., positive affect) and neural (e.g., AA) indicators provides a more comprehensive framework for examining how individual differences in brain function and affect relate to naturally occurring PA patterns.

The present study aimed to determine whether resting AA predicts free-living PA patterns. Using accelerometer-derived metrics that capture both PA volume and intensity, we examined how frontal and parietal AA relate to natural variations in PA behavior. A secondary aim was to evaluate whether positive affect contributes to or strengthens these associations, reflecting the potential interplay between stable neural dispositions and affective states in shaping real-world activity engagement. Based on prior evidence linking approach motivation with less left hemisphere power, we hypothesize that (i) lower AA power in the left frontal and left parietal regions (reflecting greater cortical activation) would predict higher PA volume and intensity, (ii) higher AA power in the right frontal and right parietal regions (reflecting lower cortical activation) would predict greater sedentary time (ST), and (iii) positive affect would independently contribute to the prediction of all PA outcomes (e.g., AvAcc, IG, MX). By integrating accelerometry, AA, and affective measures, this study addresses a critical gap in understanding how neural and psychological mechanisms interact to shape free-living PA behavior.

## 2. Materials and Methods

### 2.1. Post Hoc Power Analysis

A post hoc power analysis was conducted using GPower (version 3.1.9.7; Heinrich Heine University Düsseldorf, Düsseldorf, Germany) [51] to determine the achieved power based on the sample size, effect size, and alpha level. The effect size ($f^{2}$) was calculated using the formula $f^{2}=\frac{R^{2}}{1-R^{2}}$ where R^2^ = 0.30. Substituting the observed value, the $f^{2}$ was 0.43. The power analysis was performed for a multiple linear regression model with three predictors (e.g., sex, positive affect, and AA) using an alpha level of 0.05 and a total sample size of 59 participants. The analysis specified an F-test for a linear multiple regression model with a fixed model and R^2^ deviation from zero as the statistical test. The achieved power, calculated using these parameters, was 0.99, with a critical F-value of 2.77. These results suggest that the study was adequately powered to detect meaningful relationships within the regression model.

### 2.2. Participants

One hundred seventeen young adults were recruited via flyers and advertisements in undergraduate and graduate Kinesiology classes at the University of North Carolina at Greensboro (UNCG). Inclusion criteria were: age between 18 and 30 years old, normal or corrected-to-normal vision (minimum 20/20 standard), no health risk for engaging in moderate or high-intensity PA (based on PAR-Q), and absence of neurological conditions that could hinder session completion. Of the 117 individuals who consented, 59 remained in the final analytic sample after accounting for missed visits, missing EEG or accelerometer data, missing PANAS responses, and outlier removal. Figure 1 provides a detailed flow diagram illustrating participant progression from initial recruitment to the final analytic sample. After exclusions, 59 participants (mean age = 21.76 ± 2.92 years; gender: 42 females and 17 males; race: 64% white or Caucasian; 20% black or African American; 5% Asian; 3% Hispanic; 2% American Indian or Alaska Native; 2% Latino; 2% Mexican; 2% Not reported) were included in the study. Participants provided written informed consent via digital signature using an online Qualtrics survey (Qualtrics, Provo, UT, USA). This study was approved by the University of North Carolina at Greensboro’s Institutional Review Board (IRB). Participants who completed the study were entered into a drawing for a $20 gift card.

### 2.3. Measures

#### 2.3.1. Accelerometry

Free-living PA was measured using the GT9X Link accelerometer (Actigraph, Pensacola, FL, USA). Participants wore the accelerometer on their non-dominant wrist for seven consecutive days during waking hours. They were instructed to remove it only for water-based activities such as swimming or showering. The devices were initialized to start recording as soon as the participant arrived at the laboratory, but they started wearing them once the appointment was finished. The sampling rate was set at 30 Hz.

Accelerometry data were downloaded using Actilife (v6.13.4; Actigraph, Pensacola, FL, USA), saved in raw format (.gt3x files), and converted to .csv format for signal processing. Data were processed in RStudio (v. 2024.04.2; Posit Software, PBC, Boston, MA, USA) [52] using the GGIR package (v3.1–5) [53]. The acceleration signals were auto-calibrated using local gravity as a reference [54] and expressed into gravity-corrected vector magnitude units (Euclidean norm minus one, ENMO, [55]). The average ENMO values were calculated over 5 s epochs and expressed in milli-gravitational units (mg) (a unit of acceleration); therefore, a higher value in any metric indicates greater intensity. Accelerometer files were excluded if post-calibration error exceeded 0.01 g (10 mg) or wear data were missing for any 15 min period of the 24 h cycle. Non-wear time was identified using Van Hees’ non-wear algorithm, which detects periods of sustained low acceleration variability by evaluating the standard deviation and value range of each axis within 60 min windows, advancing in 15 min increments. Non-wear was classified when the standard deviation was <13 mg or the value range was <50 mg for at least two of the three axes, following GGIR defaults [55]. Participants required at least one valid day wear [56], defined as at least 10 h [57], to be included. Hildebrand’s cut-points [58,59] for the non-dominant wrist were applied to classify time spent in LPA (44.8 mg), moderate PA (100.6 mg), and vigorous PA (428.8 mg). Metrics such as AvAcc, IG, and the MX metrics were derived, specifically the most active minutes across 120, 60, 30, 15, 10, 5, and 2 min (M120, M60, M30, M15, M10, M5, and M2, respectively) and interpreted as described by Rowlands et al. [50]. Appendix A maps GGIR variable names to their respective nomenclature.

#### 2.3.2. EEG

##### Recording

EEG activity was recorded using a 64-electrode Neuroscan Quick-Cap (Compumedics, Charlotte, NC, USA), following the international 10–10 systems [60]. Electrooculographic (EOG) signals were recorded to account for eye movements and blinks using electrodes placed at the outer canthus of each eye and above and below the left orbit. Signals were referenced online to a midline electrode between Cz and CPz, with Fz acting as the ground electrode. Data were digitized at a sampling rate of 1000 Hz, amplified 500 times, and filtered with a DC to 70 Hz band pass filter—to record desired neural activity—and a 60 Hz notch filter—to reduce powerline noise—using a Neuroscan SynAmps2 amplifier (Compumedics, Charlotte, NC, USA). Impedance values were maintained below 10 kΩ. Recordings of the resting state occurred over a 4 min period with eyes closed. AA has been shown to be more stable when the eyes are closed compared to when they are open [25,61,62].

##### Processing

Offline data was processed utilizing MATLAB (R2024a) and various toolbox plug-ins, including EEGLAB [63] and ERPLAB [64]. Data were re-referenced to the average mastoids (M1, M2) and high-pass filtered at 0.1 Hz. Bad channels were cleaned and/or temporally removed using artifact subspace reconstruction (ASR) [65,66] on a channel-by-channel basis with the following settings: cutoff threshold of 4 standard deviations; window length was set to 0.5 s; step size of 0.1 s, and the maximum number of dimensions retained for the signal subspace was set to 10. ASR was chosen because it effectively attenuates brief, non-stationary artifacts (e.g., muscle bursts or electrode shifts), allowing for more accurate preservation of the underlying resting-state signal [67]. Eyeblink artifacts were removed using an automated independent component analysis (ICA) procedure with the extended infomax algorithm. Following the ICA computation, eyeblink artifact components were identified using the icablinkmetrics plugin [68] based on correlations (r = 0.80) comparing the input artifact channel (i.e., VEOG) with the ICA waveform. Removed channels were interpolated using spherical spline interpolation to maintain dataset integrity.

##### AA

For AA, a low-pass filter with a cutoff frequency of 50 Hz was applied, and a DC offset correction was performed. Events were marked at 2 s intervals to segment the continuous EEG data into epochs for spectral analysis. Epochs were extracted from −1 to 1 s relative to the event markers, and baseline correction was applied using the −100 to 0 ms pre-event period. Power spectral analysis was conducted for the EEG channel using the spectopo function, which computes the power spectrum in decibels (dB) across the frequency range of interest. For the alpha band (8–13 Hz), spectral power values were extracted based on the corresponding frequency indices. Absolute power (µV^2^) in the alpha band was calculated by converting the dB values to linear power (10^dB/10^) and averaging across the specified frequency range. AA was created by subtracting the natural log-transformed right hemisphere power from the left hemisphere power (e.g., log_10_[P4] − log_10_[P3] = AA). Specific electrodes pairs were chosen from the frontal (FP2 − FP1, AF4 − AF3, F8 − F7, F6 − F5, F4 − F3, F2 − F1, FT8 − FT7) and the parietal region (P8 − P7, P6 − P5, P4 − P3, P2 − P1). Because alpha power is inversely related to cortical activation [18,69], lower AA power values reflect greater hemisphere activation in that same region. For example, if the subtraction log_10_[P4] − log_10_[P3] yields a positive value, the right hemisphere (P4) shows higher alpha power than the left (P3), indicating lower activation of the right hemisphere’s cortex, while simultaneously suggesting greater activation of the left hemisphere’s cortex.

#### 2.3.3. The Positive and Negative Affect Schedule (PANAS)

The PANAS is a 20-item questionnaire with 10 items measuring positive affect and 10 items measuring negative affect [70]. Participants were instructed to rate the extent to which they experience each emotion right now using a 5-point Likert Scale, ranging from 1 = Very Slightly or Not at all to 5 = Extremely. Positive affect scores were calculated from items 1, 3, 5, 9, 10, 12, 14, 16, 17, and 19; negative affect scores were calculated from items 2, 4, 6, 7, 8, 11, 13, 15, 18, and 20.

### 2.4. Procedure

Participants attended the laboratory on two different occasions, one week apart. On the first day, they completed the PANAS questionnaire using Qualtrics. Afterward, participants were fitted with the EEG cap and underwent a 4 min continuous recording with their eyes closed. At the end of the visit, participants were instructed to wear the activity monitor on their wrist for seven consecutive days during waking hours. On their second visit, they returned the device.

### 2.5. Statistical Analysis

Descriptive statistics (means and standard deviations) summarized participant characteristics. Outliers in AA and accelerometer-derived outcomes were excluded if values exceeded ±3 standard deviations from the mean (See Table 1). A correlation matrix was first generated to understand possible associations between variables (Appendix A). For subsequent analysis, only positive affect was included in the model. This decision was based on prior research demonstrating that positive affect is more consistently linked to the initiation and maintenance of PA than negative affect, which tends to show weaker to no associations overall [71,72]. Accordingly, focusing on positive affect provides a more reliable indicator of affective processes that promote real-world behavior. Multiple linear regression models examined the relationship between AA (predictor) and PA outcomes (i.e., LPA, MVPA, AvAcc, IG, and MX metrics) and ST, controlling for sex (0 = female, 1 = male) and positive affect. AA was operationalized using individual electrode pairs (e.g., FP2 − FP1, P4 − P3). Standardized β coefficients and R^2^ values quantified the influence of the predictors. To control for type I error due to multiple comparisons, the Benjamini–Hochberg False Discovery Rate (FDR) procedure [73] was applied across 11 planned models (*m*). Individual *p*-values were ranked (*i*) in ascending order, and critical values were computed as (*i*/*m*) × 0.15 [74]. Analyses were performed using IBM SPSS Statistics (v30.0; IBM Corp., Armonk, NY, USA), with α = 0.05.

## 3. Results

For clarity, only significant regression models are reported in the results, with all regression model outcomes reported in the Appendix A for transparency. Only significant main effects are reported after FDR correction.

Table 1 presents the sample characteristics. Participants were generally overweight and predominantly non-Hispanic, and the sample consisted mostly of females. Participants accumulated more ST compared to LPA and MVPA. On average, IG and AvAcc suggested relatively low overall activity. The MX metrics (M120 to M2) indicated that the highest-intensity minutes of the day were more often composed of shorter or sporadic activity segments (e.g., 2–15 min), rather than longer continuous periods. Affective responses showed moderately high positive affect and relatively low negative affect.

### Frontal and Parietal AA as Predictors of PA

No significant effects were observed for regression models including ST, LPA, MVPA, AvACC, M120, and M2 from any electrode pair (Appendix A).

For IG, the overall model was significant for P2 − P1 (see Table 2, Appendix A and Figure 2). Higher positive affect was associated with greater IG (*p* < 0.01), indicating greater engagement in higher-intensity activities throughout the day. AA was significant at P2 − P1 (*p* < 0.05). The negative β coefficient indicates that greater left-hemisphere alpha power (i.e., lower cortical activation) was associated with lower IG, suggesting reduced engagement in higher-intensity PA across the day. However, AA effects did not remain significant after FDR correction, and no significant sex effect was observed.

For M60, the overall model was significant (see Table 2, Appendix A and Figure 2). Higher positive affect was associated with higher M60 values (*p* < 0.001), indicating greater PA levels during their most active 60 min of the day. AA was significant at P6 − P5 (*p* < 0.05). The positive β coefficient indicates that greater right-hemisphere alpha power (i.e., lower cortical activation) was associated with higher M60, suggesting that individuals tended to accumulate more PA during their most active 60 min. However, AA effects did not remain significant after FDR correction. No significant sex effect was observed.

For M30, overall significant models were found for P6 − P5 and P4 − P3 (see Table 2, Appendix A and Figure 2). Higher positive affect was associated with higher M30 values (P6 − P5: *p* < 0.001; P4 − P3: *p* < 0.001), indicating greater PA levels during their most active 30 min of the day. Additionally, AA at both sites was significant (P6 − P5: *p* < 0.05; P4 − P3: *p* < 0.05). The positive β coefficient indicates that greater right-hemisphere alpha power (i.e., lower cortical activation) was associated with higher M30, suggesting that individuals tended to accumulate more PA during their most active 30 min. These electrodes remained significant after FDR correction. Sex trended toward significance in both models (P6 − P5: *p* = 0.067; P4 − P3: *p* = 0.059).

For M15, the overall models for P6 − P5 and P4 − P3 were significant (see Table 2, Appendix A and Figure 2). Higher positive affect was associated with higher M15 values (P6 − P5: *p* < 0.001; P4 − P3: *p* < 0.001), indicating greater PA levels during their most active 15 min of the day. AA was also significant at both sites (P6 − P5: *p* < 0.05; P4 − P3: *p* < 0.05). The positive β coefficient indicates that greater right-hemisphere alpha power (i.e., lower cortical activation) was associated with higher M15, reflecting greater volume PA accumulation during their most active 15 min of the day. These electrodes remained significant after FDR correction. Additionally, sex emerged as a significant predictor in both models (P6 − P5: *p* = 0.044; P4 − P3: *p* = 0.038), indicating that males accumulated greater PA volume during their most active 15 min of the day.

For M10, significant models were observed for P6 − P5 and P4 − P3 (see Table 2, Appendix A and Figure 2). Higher positive affect was associated with higher M10 values (P6 − P5: *p* < 0.001; P4 − P3: *p* < 0.001), indicating greater PA levels during their most active 10 min of the day. AA was also significant at both sites (P6 − P5: *p* < 0.01; P4 − P3: *p* < 0.05). The positive β coefficient indicates that greater right-hemisphere alpha power (i.e., lower cortical activation) was associated with higher M10, suggesting that individuals tended to accumulate more PA during their most active 10 min. However, AA effects did not remain significant after FDR correction. Additionally, sex emerged as a significant predictor in both models (P6 − P5: *p* = 0.044; P4 − P3: *p* = 0.038), indicating that males accumulated greater PA volume during their most active 10 min of the day.

For M5, significant models emerged for P6 − P5 and P4 − P3 (see Table 2, Appendix A and Figure 2). Higher positive affect was associated with higher M5 values (P6 − P5: *p* = 0.001; P4 − P3: *p* = 0.002), indicating greater PA levels during their most active 5 min of the day. AA was also significant at both sites (P6 − P5: *p* = 0.005; P4 − P3: *p* = 0.020). The positive β coefficient indicates that greater right-hemisphere alpha power (i.e., lower cortical activation) was associated with higher M5, suggesting that individuals tended to accumulate more PA during their most active 5 min. However, AA effects did not remain significant after FDR correction, and no significant effect of sex was observed.

## 4. Discussion

The purpose of this study was to examine the relationship between AA power in the frontal and parietal regions and accelerometer-derived PA metrics—including total volume, intensity, and the most active periods of the day (MX). Our first hypothesis tested whether lower AA power in the left frontal and left parietal regions (reflecting greater cortical activation) would predict higher PA volume and intensity. The results showed that this hypothesis was partially supported. After applying FDR correction, AA effects remained significant only for the M30 and M15 models, both within the right parietal region. Specifically, more rightward parietal AA (i.e., greater relative right-than-left alpha activity) was associated with higher activity during participants’ most active 30 and 15 min of the day. For the IG, the relationship was stronger, with greater AA power in the left parietal hemisphere than in the right. No significant associations were found for LPA, MVPA, AvAcc, M120, or M2. Additionally, none of the models revealed significant effects in the frontal regions. Our second hypothesis stated that higher AA power in the right frontal and right parietal regions (reflecting lower cortical activation) would predict greater ST. Our findings did not provide any significant association between ST and AA power. Lastly, our third hypothesis proposed that positive affect would independently contribute to the prediction of all PA outcomes (e.g., AvAcc, IG, MX). As expected, positive affect emerged as a consistent predictor of all PA behaviors.

Most of the existing evidence in AA, PA, and exercise stems from crossover [17,75] and cross-sectional studies [32,33]. These studies consistently reported that greater PA is typically associated with lower left frontal AA power (i.e., greater cortical activation), whereas higher ST is associated with increased left frontal AA power (i.e., lower cortical activation). Our results diverged from this reduced left frontal AA power linked to greater PA, demonstrating that greater AA power in the right parietal hemisphere (i.e., lower cortical activation) predicted PA only during the most active 30 and 15 min periods. These converging findings could be due to over-reliance on self-reported PA measurements rather than objective measurements (i.e., accelerometer used in this study), which introduces limitations [34,35]. Indeed, while participants in those studies reported <6 h of ST per day, our accelerometer-based data indicated an average of 12 h. The population estimate for sitting time is about 9 h daily [76]. Thus, discrepancies across studies likely reflect methodological differences in PA assessment that could impact brain outcomes such as AA in this case. Importantly, traditional accelerometer cut-points such as MVPA and LPA collapse heterogeneous movement patterns into broad categories and are affected by calibration and population-specific biases [44]. In contrast, metrics like the most active X minutes (MX) quantify how activity is accumulated across discrete time intervals, providing a temporally resolved representation of peak movement behavior [49,50]. This distinction is critical for AA research, as our findings show that neural–behavioral associations emerged only for M30 and M15 (i.e., metrics that capture sustained, high-engagement activity windows). In contrast, traditional cut-point metrics (MVPA/LPA) did not relate to AA.

Aerobic exercise paradigms have repeatedly shown reduced left frontal AA power post-exercise (i.e., greater cortical activation), with effects modulated by both intensity and duration [17,24,25,26,77,78]. In fact, exercise sessions lasting 30 min and approximately 70%VO_2_ max have been associated with enhanced right frontal AA power [17,25,26]. Although this literature highlights hemispheric lateralization during exercise, our results did not reveal any frontal AA associations with PA. Instead, AA power and PA associations emerged only in the right parietal region [19,21,24]. In particular, higher right parietal alpha power was associated with greater activity during the most intense portions of the day, with preliminary effects observed for activity accumulated across both longer, lower-intensity windows (e.g., M60 − M30) and shorter, higher-intensity periods (e.g., M15 − M5) [50]. However, after applying FDR correction, these effects were limited to activity accumulated during the most active 30 and 15 min of the day (M30 − M15), a pattern that conceptually aligns with laboratory studies using 30 min moderate-intensity protocols [17,25,26].

When examining the accumulation of intensity across the day (IG), the relationship with this and AA power was greater in the left parietal hemisphere (i.e., lower cortical activation) compared to the right parietal hemisphere (i.e., greater cortical activation), even though it did not survive FDR. Prior work suggests that increasing exercise intensity engages affective and reward-related neural systems, particularly within the right anterior insula and associated interoceptive networks [37,38,79], which supports, among others, emotional salience and stimulus valence. Together, these processes may function as an alarm system, signaling heightened interoceptive awareness before engaging in activities that represent a substantial homeostatic challenge. Such neural responses could serve to prevent or delay participation in strenuous bouts. These concepts may align with previous evidence from right AA power, in which a lower power (as shown in our results) has been connected with withdrawal tendencies and aversive affective states [12,19,20,21,22,23].

While most mechanistic explanations of AA and PA associations emphasize frontal regions, our findings suggest that parietal contributions warrant greater consideration. The parietal cortex is critically involved in multisensory integration, visuospatial processing, and motor planning [80,81,82]. All these functions are directly recruited when coordinating whole-body movement in free-living activity. Thus, parietal AA may be a more proximal neural marker of movement readiness and sensorimotor coupling than frontal AA, which more strongly reflects affective and motivational biasing rather than moment-to-moment motor engagement [12,19,20,21,22,23]. Moreover, posterior parietal and fronto-parietal networks contribute to interoceptive monitoring and attentional allocation to bodily signals [37], providing a plausible mechanism linking parietal AA to decisions about initiating or sustaining physical activity. Neurotransmitter pathways such as serotonin, dopamine, and noradrenaline project broadly across fronto-parietal circuits [83], indicating that parietal AA may reflect system-level neuromodulatory changes accompanying physical exertion. Supporting this view, recent work demonstrates reductions in parietal AA following graded exercise to exhaustion [27], highlighting the sensitivity of posterior regions to metabolic demand. One plausible explanation relates to the inverse relationship between alpha power and cortical metabolic activity [84]. Given the brain’s limited metabolic capacity [85,86], sustained prefrontal engagement—particularly in the dorsolateral and ventrolateral prefrontal cortex—may require a redistribution of energetic resources away from posterior regions such as the motor-parietal cortex [87]. Within this framework, our observation of heightened right parietal alpha during intensity periods over time may represent an adaptive inhibitory mechanism that downregulates sensory input to preserve motor efficiency.

In contrast, no significant associations were found between frontal AA and ST, LPA, MVPA, AvAcc, M120, and M2. These null effects highlight a key methodological consideration—broad summary and traditional measures of PA (e.g., LPA and MVPA) may obscure the rapid fluctuations in affect, effort, and motivational engagement that AA is theorized to track [14,88]. Similarly, global movement indicators like AvAcc and the M2 metric index overall activity volume or the single most active 2 min of the day, which may not capture the sustained periods of coordinated motor output and affective–motivational engagement required to elicit reliable AA and PA associations [26,50]. Together, these considerations may help explain the consistent relationship between AA and specific temporally Mx metrics—particularly M30 and M15—which better represent extended periods of active engagement and are aligned with the contemporary 24 h movement behavior framework, emphasizing the dynamic patterning of activity across the day [89,90].

This study offers several notable strengths. First, it integrates multimodal data—combining EEG-derived AA, accelerometer-based PA metrics, and affective self-report—to provide a comprehensive understanding of the neural and psychological correlates of free-living PA. The use of objective accelerometry data enhances ecological validity and avoids the recall biases common in self-reported PA measures. Additionally, the application of novel accelerometry metrics such as IG, AvAcc, and the most active X minutes (MX) allows for a more nuanced characterization of PA patterns beyond traditional cut-points (ST, LPA, and MVPA). A key contribution of this work is its focus on parietal AA, an underexplored region in the context of PA and affect, which yielded novel insights into the neural correlates of high-intensity, short-duration activity.

Despite these strengths, several limitations should be acknowledged. The cross-sectional design precludes causal inference, limiting the ability to determine directional relationships between AA, PA, and affect. The sample was predominantly female, which may affect the generalizability of the findings and potentially influence AA patterns. Additionally, affect was measured only once, preventing alignment with daily PA fluctuations and limiting temporal resolution. Finally, while the sample size was sufficiently powered for primary analyses, it may have been underpowered to detect interaction effects, particularly across multiple electrode pairs and PA metrics.

## 5. Conclusions

This study provides novel evidence that parietal AA, rather than frontal AA, may be more relevant for understanding how individuals accumulate activity during their most intense periods of the day. More broadly, these findings suggest that motivational and affective processes reflected in parietal AA could influence when and how people choose to move in free-living environments. Additionally, the strong sensitivity of the M30 and M15 metrics underscores the importance of using time-specific accelerometer indicators to detect neural–behavioral relationships that traditional PA categories (LPA and MVPA) may miss. Identifying neurophysiological markers linked to naturally occurring patterns of peak activity has the potential to improve personalized PA interventions by tailoring strategies to individuals’ affective-motivational profiles. Future work should examine whether parietal AA predicts changes over time, whether it can be modified through behavioral or exercise-based interventions, and how these relationships operate across different populations and contexts.

## Figures and Tables

**Figure 1 brainsci-15-01322-f001:**
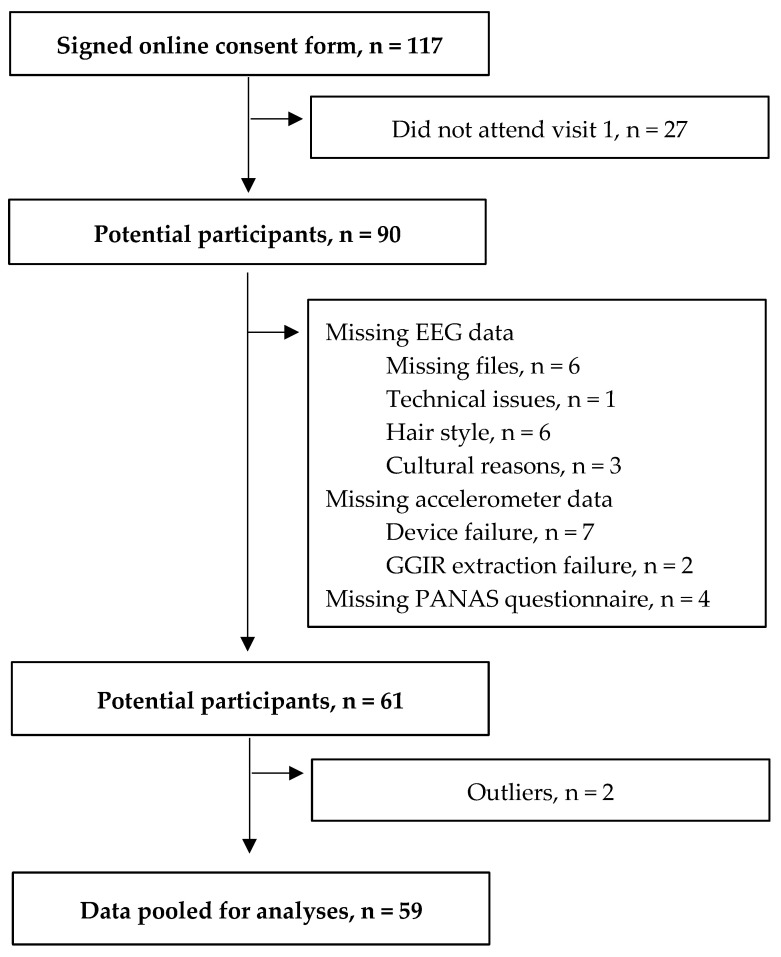
Flow diagram for study participants. Note. Positive and Negative Affect Schedule (PANAS).

**Figure 2 brainsci-15-01322-f002:**
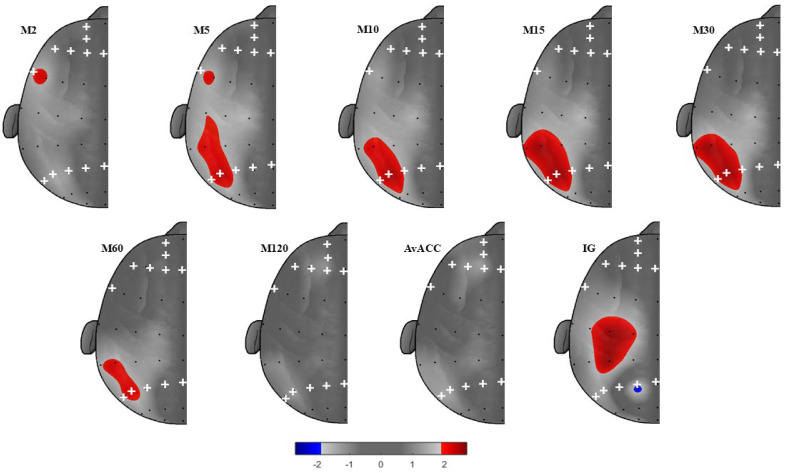
Topographic plots for alpha power representing all PA behaviors. Note: Maps represent *t*-values from regression models of contralateral–ipsilateral alpha power. Red colors indicate higher *t*-values for right alpha power, representing stronger associations with AA and PA behaviors. Blue colors indicate higher *t*-values for left alpha power, representing weaker associations of AA with PA behaviors. White plus symbols (+) indicate electrodes included in the analyses (FP1 − FP2/AF4 − AF3/F8 − F7/F6 − F5/F4 − F3/FT8 − FT7/P8 − P7/P6 − P5/P4 − P3/P2 − P1).

**Table 1 brainsci-15-01322-t001:** Demographics of the sample.

Measure	Mean (±SD)
Age (years)	21.76 (2.92)
Height (cm)	170.49 (10.50)
Weight (kg)	76.36 (15.92)
BMI (kg/m^2^)	26.18 (5.00)
Gender (n [%])	
Male	17 [29%]
Female	42 [71%]
Ethnicity (n [%])	
Non-Hispanic	43 [73%]
Hispanic	12 [20%]
No response	4 [7%]
ST (min/day)	743.32 (95.91)
LPA (min/day)	142.00 (35.36)
MVPA (min/day)	109.34 (36.52)
IG (mg)	−2.48 (0.16)
AvAcc (mg)	28.82 (6.94)
M120 (mg)	69.78 (21.32)
M60 (mg)	89.36 (30.49)
M30 (mg)	110.78 (40.75)
M15 (mg)	135.72 (52.34)
M10 (mg)	152.76 (63.26)
M5 (mg)	183.50 (81.80)
M2 (mg)	235.41 (136.81)
^a^ Positive affect	24.90 (9.62)
^a^ Negative affect	12.98 (3.09)

Abbreviations: BMI, Body mass index; milli-gravitational units (mg); ST, sedentary time; LPA, light physical activity; MVPA, moderate-to-vigorous physical activity; IG, intensity gradient; AvAcc, average acceleration, M2; the most active 2 min of the day; M5, the most active 5 min of the day; M10, the most active 10 min of the day; M15, the most active 15 min of the day; M30, the most active 30 min of the day; M60, the most active 60 min of the day; M120, the most active 120 min of the day. Note: height, weight, and BMI were only reported with data from 53 participants. ^a^ Measures obtained from the Positive and Negative Affect Schedule (PANAS).

**Table 2 brainsci-15-01322-t002:** Regression model summary table of AA power predicting PA, accounting for positive affect and sex.

Overall Model for Sex, Affect, and EEG				Predictor					
	R^2^	F	*p*	B	B_error_	β	t	*p*	CI
**IG**									
P2 − P1	0.264	6.573	0.001	−0.403	0.185	−0.264	−2.180	0.034 *	−0.774, −0.033
**M60**									
P6 − P5	0.234	5.614	0.002	34.700	14.600	0.292	2.377	0.021 *	5.441, 63.960
**M30**									
P6 − P5	0.295	7.669	0.000	48.924	18.724	0.308	2.613	0.012 *	11.401, 86.448
P4 − P3	0.276	6.976	0.000	60.759	26.699	0.268	2.276	0.027 *	7.254, 114.264
**M15**									
P6 − P5	0.297	7.761	0.000	62.783	24.009	0.308	2.615	0.011 *	14.668, 110.897
P4 − P3	0.278	7.076	0.000	78.156	34.227	0.269	2.283	0.026 *	9.563, 146.748
**M10**									
P6 − P5	0.265	6.606	0.001	73.913	29.679	0.300	2.490	0.016 *	14.434, 133.391
P4 − P3	0.250	6.105	0.001	94.063	42.178	0.268	2.230	0.030 *	9.538, 178.589
**M5**									
P6 − P5	0.260	6.440	0.001	91.054	38.508	0.286	2.365	0.022 *	13.882, 168.227
P4 − P3	0.244	5.918	0.001	113.726	54.752	0.250	2.077	0.042 *	4.000, 223.453

Note: Neither ST, LPA, MVPA, AvACC, M120, nor M2 presented significant electrodes. Abbreviations: IG, Intensity gradient; M60, most active 60 min; M30, most active 30 min; M15, most active 15 min; M10, most active 10 min; M5, most active 5 min; R^2^, R square; B, unstandardized beta coefficients; B_error_, unstandardized standard error; CI, Confidence interval 95%. * *p* < 0.05.

## Data Availability

The data presented in this study are available on request from the corresponding author due to privacy and ethical considerations.

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
