# Peer review of "The Relation of Alpha Asymmetry to Physical Activity Duration and Intensity"

_brainsci, 2025, doi:10.3390/brainsci15121322_

Round 1
Reviewer 1 Report
Comments and Suggestions for Authors
Dear Authors,
Thank you for providing me with the opportunity to read thsi interesting paper. Below, I have listed my comments:
1) The introduction is thorough, well-cited, and demonstrates strong knowledge of the PA + EEG + affective neuroscience literature. There is a minor clarification i would ask for. The link between AA and PA is sometimes implied but not explained deeply (e.g., why approach/avoidance is relevant to sustained PA).
2) Before presenting your aims, you could explicitly state what is missing e.g. lack of multi-regional (frontal + parietal) AA studies, limited research in younger adults, reliance on self-report in prior work, or limited integration of affect + neural + objective PA.
3) In the methods, Accelerometry and EEG processing contain extensive technical parameters, while PANAS and procedure sections are relatively brief. This creates an imbalance.
4) In my opinion, the rationale for including only positive affect suits better in the Analysis section or a brief conceptual justification could be stated, but not in Measures.
5) It is best to state how sample size dropped from 117 to 59 in one concise sentence before the diagram.
6) A sentence explaining why ASR was chosen would help clarity.
7) In the discussion, it will be benefitial to clarify the directionality of AA effects. Multiple statements reference 'greater AA power' but readers may be confused given that higher alpha power implies lower cortical activation. Explicit reminders would help avoid misinterpretation.
8) rationale for why parietal AA. but not frontal AA, was associated with PA:
9) Your interpretation is good but scattered across paragraphs. A more focused explanation of parietal roles in sensorimotor integration, movement readiness. attentional engagement, and interoception would sharpen the narrative.
10) After all the detailed interpretation, the Discussion needs a more explicit, concise takeaway message. Your finding that M30/M15 are more predictive than MVPA/LPA is important and could be emphasized more.
I hope this feedbakc is helpful.
Author Response
Thank you for providing me with the opportunity to read this interesting paper.
We appreciate the reviewer’s positive assessment and are grateful for the opportunity to strengthen the manuscript.
1) The introduction is thorough, well-cited, and demonstrates strong knowledge of the PA + EEG + affective neuroscience literature. There is a minor clarification I would ask for. The link between AA and PA is sometimes implied but not explained deeply (e.g., why approach/avoidance is relevant to sustained PA).
We thank the reviewer for nothing the need to more clearly articulate why approach/avoidance tendencies are theoretically relevant for sustained physical activity. We have revised the Introduction and explicitly explain how motivational orientation indexed by AA may shape behavioral engagement patterns in real-world movement. This comment was addressed in the main document. See page 2, lines 62–65.
2) Before presenting your aims, you could explicitly state what is missing e.g. lack of multi-regional (frontal + parietal) AA studies, limited research in younger adults, reliance on self-report in prior work, or limited integration of affect + neural + objective PA.
We agree that a more explicit articulation of the missing gaps strengthens the transition to the study aims. The Introduction now clearly outlines the absence of multi-regional (frontal + parietal) AA studies, the limited evidence in younger adults, and the heavy reliance on self-report in prior work. See page 3, lines 118–120.
3) In the methods, Accelerometry and EEG processing contain extensive technical parameters, while PANAS and procedure sections are relatively brief. This creates an imbalance.
We appreciate the reviewer’s observation. The more extensive reporting of EEG and accelerometer procedures reflects the level of technical detail required for transparency and replicability in these modalities. In contrast, the PANAS is a widely established measure with well-validated administration procedures, which is why its description is necessarily more concise. Similarly, the procedural section is brief because the study employed a straightforward cross-sectional design with minimal experimental manipulation. For these reasons, we maintained the current level of detail to preserve methodological clarity without redundancy.
4) In my opinion, the rationale for including only positive affect suits better in the Analysis section or a brief conceptual justification could be stated, but not in Measures.
We agree that the justification fits more naturally within the Analysis section. We relocated this paragraph accordingly. See page 7, lines 290–294.
5) It is best to state how sample size dropped from 117 to 59 in one concise sentence before the diagram.
A concise explanation for the reduction in sample size has been added to the Participants section, along with a reference to the flow diagram that details the exclusion process. See page 4, lines 161–165.
6) A sentence explaining why ASR was chosen would help clarity.
We added a brief justification describing why ASR was selected, emphasizing its strengths for mitigating movement-related and non-stationary artifacts typical of ambulatory EEG. See page 6, lines 238–240.
7) In the discussion, it will be beneficial to clarify the directionality of AA effects. Multiple statements reference 'greater AA power' but readers may be confused given that higher alpha power implies lower cortical activation. Explicit reminders would help avoid misinterpretation.
We appreciate this important point. The Discussion has been revised to explicitly remind readers that higher alpha power indexes lower cortical activation, especially when referring to “greater AA power.”
8) Rationale for why parietal AA. but not frontal AA, was associated with PA & 9) Your interpretation is good, but scattered across paragraphs. A more focused explanation of parietal roles in sensorimotor integration, movement readiness. attentional engagement, and interoception would sharpen the narrative.
We addressed these two related comments together. The revised Discussion provides a more focused explanation of parietal cortex functions—sensorimotor integration, attentional engagement, movement readiness, and interoception—and articulates why AA in parietal regions may be more sensitive to real-world movement tendencies than frontal AA. Supporting literature has been added to strengthen this rationale. See pages 13, lines 464–472.
10) After all the detailed interpretation, the Discussion needs a more explicit, concise takeaway message. Your finding that M30/M15 are more predictive than MVPA/LPA is important and could be emphasized more.
We agree that the Discussion benefits from a clearer summary of the key conceptual contributions, particularly the stronger predictive value of M30/M15 metrics over traditional MVPA/LPA variables. This emphasis has been strengthened in both the Discussion and the Conclusion. See pages 12–13, lines 427–435 and 520–523.
Reviewer 2 Report
Comments and Suggestions for Authors
Please see the attached file

Author Response
The manuscript is well written, methodologically sound, and addresses a topical issue: the association between alpha asymmetry (AA), positive affect, and physical activity in real-life conditions.
We thank the reviewer for these encouraging comments and for the insightful suggestions that improved the clarity and coherence of the manuscript.
The abstract is well structured, but could be improved in the results section by motivating and strengthening the theoretical justification for the shift from frontal AA to parietal AA as the primary neural correlate. The conclusions focus mainly on parietal AA, but do not fully address the role of frontal AA, which was mentioned in the objective. It is therefore suggested that the results be linked back to the broader neural context introduced (frontal vs. parietal) to justify the focus on in the results section only on the parietal AA
We appreciate this suggestion. The abstract has been revised to more clearly indicate why the interpretation focuses on parietal AA, linking this shift back to the broader neural context introduced in the paper. These changes were made while maintaining the 250-word limit. See page 4, lines 21–22 and 26–29.
Introduction
2) The introduction is well constructed and rich in recent citations. However, some sections (in particular the lengthy explanation of accelerometric metrics) could be streamlined or moved to the Methods section to make for smoother reading. The conceptual background on AA is clear, comprehensive, and consistent with the study objectives.
We appreciate the comment regarding the metrics description; however, we consider that an extended description of these metrics is needed, as they represent emerging accelerometer-based indicators that are increasingly used in the health literature but remain unfamiliar in the fields of cognition, affect, and brain. The advantage of these metrics is that they go beyond light PA and moderate-to-vigorous PA, as they capture different dimensions of movement behavior, which makes their interpretation meaningful for understanding the study’s hypotheses and later results. Moving these descriptions to the Methods section would disconnect the conceptual significance of these metrics from the theoretical framework presented in the Introduction. Keeping them in the Introduction provides necessary context, ensures conceptual transparency, and helps readers understand why these metrics meaningfully advance prior work beyond traditional cut-points.
The hypotheses are well defined. However, it would be useful for the authors to include a brief summary in the Discussion clarifying which hypotheses were confirmed and which were not, in order to facilitate understanding of the results.
We agree this improves clarity. A brief restatement summarizing which hypotheses were supported appears in the revised Discussion. See page 12, lines 396–413.
Materials and methods
In the Participants section, you have to describe ,also in this specific Paragraph , the average age and standard deviation of the sample and the distribution by gender.
Mean age, standard deviation, and gender distribution have now been incorporated into the Participants section. See page 4, lines 165-166.
I suggest that you also specify here the main reasons why 58 participants (117-59) were excluded.
A concise explanation of why 58 participants were excluded has been added in the Participants section, with a reference to Figure 1 for a detailed flow of exclusions. See page 4, lines 161–165.
Round 2
Reviewer 1 Report
Comments and Suggestions for Authors
Thank you Authors for revising the manuscript. Good luch with the rest of the process.